# Dual Diagnosis in Adolescents with Problematic Use of Video Games: Beyond Substances

**DOI:** 10.3390/brainsci12081110

**Published:** 2022-08-21

**Authors:** Arturo Esteve, Antonio Jovani, Ana Benito, Abel Baquero, Gonzalo Haro, Francesc Rodríguez-Ruiz

**Affiliations:** 1TXP Research Group, Universidad Cardenal Herrera-CEU, CEU Universities, 12006 Castelló, Spain; 2Mental Health Department, Consorcio Hospitalario Provincial of Castelló, 12002 Castelló, Spain; 3Torrent Mental Health Unit, Hospital General Universitario of Valencia, 46014 Valencia, Spain; 4Foundation Amigó, 12006 Castelló, Spain

**Keywords:** dual diagnosis, video games, personality, addiction, parenting

## Abstract

The technological revolution has led to the birth of new diagnoses, such as gaming disorder. When any addiction, including this one, is associated with other mental disorders, it is considered a dual diagnosis. The objectives of this current work were to estimate the prevalence of dual diagnoses in the adolescent general population while also considering the problematic use of video games and substance addiction and assessing its psychosocial risk factors. Thus, we carried out a cross-sectional study with a sample of 397 adolescents; 16.4% presented problematic videogame use and 3% presented a dual diagnosis. Male gender increased the probability of both a dual diagnosis (OR [95% CI] = 7.119 [1.132, 44.785]; *p* = 0.036) and problematic video game use (OR [95% CI] = 9.85 [4.08, 23.77]; *p* < 0.001). Regarding personality, low conscientiousness, openness, and agreeableness scores were predictors of a dual diagnosis and problematic videogame use, while emotional stability predicted a dual diagnosis (OR [95% CI] = 1.116 [1.030, 1.209]; *p* = 0.008). Regarding family dynamics, low affection and communication increased both the probability of a dual diagnosis (OR [95% CI] = 0.927 [0.891, 0.965]; *p* < 0.001) and problematic video game use (OR [95% CI] = 0.968 [0.945, 0.992]; *p* = 0.009). Regarding academic performance, bad school grades increased the probability of a dual diagnosis. In summary, male gender, certain personality traits, poor communication, and poor affective family dynamics should be interpreted as red flags that indicate an increased risk of a dual diagnosis in adolescents, which could require early intervention through specific detection programs.

## 1. Introduction

Adolescence is a time that involves great changes in physical, cognitive, emotional, and social development. Excessive substance use and video game addiction among adolescents have raised social alarm bells in recent years [1], with the latter having now been officially included in the ICD-11 [2] as gaming disorder (GD). The ever-changing nature of this disease due to rapid technological evolution provides an interesting aspect of the research, which was initially recognized by DSM-5 as a condition for further research under the term of Internet gaming disorder (IGD). Nine symptoms for IGD were listed: preoccupation with Internet games, withdrawal symptoms, tolerance, unsuccessful attempts to control the participation in Internet games, loss of interest in previous entertainment, continued excessive use of games, lies about the extent of playing, play to forget about real-life problems, and loss of relationships because of excessive game playing [3]. Problematic Internet use includes diverse activities apart from videogames, such as social media use, web-streaming, and pornography buying and viewing, which were correlated with emotional dysregulation [4]. 

Some studies reported that up to 8.2% of Spanish adolescents may suffer from GD [5], while other work showed the negative consequences of their problematic use [6]. Notwithstanding, taking into account the fact that the severity of this addiction can be placed on a continuum ranging from non-addictive subjects and GD, we found those with problematic use of video games (PUVG) have prevalence ranges between 1.3 and over 50% [7,8]. This wide variability between studies could be explained by the use of diverse assessment instruments and selection bias. PUVG has been defined as an addiction-like behavior that includes experiencing: (a) a loss of control over one’s behavior, (b) conflicts with the self and with others, (c) a preoccupation with gaming, (d) the utilization of games for purposes of coping/mood modification, and (e) withdrawal symptoms [9].

Adolescents are more vulnerable to GD and PUVG than adults, resulting in potential adverse effects on an individual’s academic and professional life [10]. Some authors argued that PUVG should be better viewed as a manifestation of underlying depressive symptoms or loneliness [11]. Therefore, PUVG can also be seen as a risky behavior, as it is usually associated with impulsivity [12], social or conduct problems, bad school performance, and specific personality types [13]. The literature considers that PUVG should be conceptualized as a behavioral addiction. Pathologic gambling is the most accurate behavioral addiction, but other behaviors can produce similar momentary rewards. While the core defining concept of substance addiction is the diminished control of ingestion of the psychoactive substance, behavioral addictions focus on a lack of control over the behavior despite adverse repercussions [14,15].

The COVID-19 pandemic, lockdown, and limitation of movements imposed by authorities increased the overall use of the Internet and videogames [16]. Moreover, rates of psychopathology (depression, anxiety, post-traumatic stress symptoms) in patients with substance or behavioral addiction have increased moderately, resulting in a poor quality of life [17]. In this context, the recently described term cyberchondria, which is understood as excessive online searching for medical information, must be considered, taking into account the elevated use of the Internet [18].

The World Psychiatric Association (WPA) defines a dual diagnosis as the concurrence of an addictive pathology (behavior or substance) and at least one other mental disorder (WPA Section on Dual Disorders). At this point, the method of understanding and dealing with addictions is changing because previously only substance disorders were considered. The underlying reason is the strong correlation between playing video games and gambling due to their commonalities in clinical expression, etiology, physiology, and comorbidity with substance use disorders according to DSM-5 [19]. Furthermore, the literature shows that symptoms in patients with GD resemble addiction-specific phenomena that are comparable to those of substance-related addictions [20]. When considering only substance addiction and not behavioral addictions, the prevalence of dual diagnoses (DDs) in adolescents is approximately 23% [21]. However, despite sharing similar psychopathological phenomena [20], the exact prevalence of DDs has not yet been determined for both these addiction types. Thus, to date, very little work studying both these disorders is available in the academic literature.

Therefore, the objective of the current study was to estimate the prevalence of DDs by exploring the comorbid presence of problematic use of video games (PUVG) with substance-related problems within a population base sample of adolescents while also exploring the relationship of these factors with family dynamics, personality, academic performance, and gender.

## 2. Materials and Methods

This observational and cross-sectional study examined a sample of 397 students aged between 13 and 17 years studying at one of five different Spanish public or private schools. Before the COVID-19 pandemic, they all completed auto-questionnaires to (1) determine whether their use of video games was problematic according to the *Video-Game-Related Experiences Questionnaire* (CERV in its Spanish abbreviation) [22] and *Game Addiction Scale for Adolescents* (GASA) [23]; (2) to assess whether they had any substance addictions by using the *CRAFFT Substance Abuse Screening Instrument* [24], *Problem-Oriented Screening Instrument for Teenagers* (POSIT) [25], and *Alcohol Use Disorders Identification Test* (AUDIT) [26]; and finally, (3) to determine whether any psychopathologies were present according to the *Behavior Assessment System for Children* (BASC) [27].

Based on these questionnaires, an independent diagnostic variable emerged with which we were able to divide the sample into three groups as follows: (1) healthy adolescents, (2) those with PUVG (above the cut-off point in at least one questionnaire), and (3) adolescents with a DD (with a psychopathology and addiction to a substance, and/or PUVG, *n* = 12). Participants who were only addicted to substances (above the cut-off point in at least two questionnaires related to substances) were excluded, as shown in Figure 1. Personality was subsequently evaluated with the *Big Five Questionnaire—Children and Adolescents* (BFQ-NA in its Spanish abbreviation) [28] and family dynamics were assessed with the *TXP Parenting Questionnaire* [29].

In the initial analysis, Pearson or Spearman correlations were performed to compare the three groups (healthy, PUVG, and DD) in terms of sociodemographic, personality, and parenting variables. Given that the variables correlated with each other, chi-squared or MANOVA tests were used to differentiate the groups, followed by ANOVA according to the type of dependent variable analyzed. To evaluate the relationships between the variables studied, we completed the analysis with an adjusted main effects multinomial regression followed by unadjusted individual multinomial regressions to predict the groups the participants belonged to with sociodemographic, personality, and parenting variables in which there were significant differences. SPSS software (version 23.0; IBM Corp., Armonk, NY) was used for all the data analyses. The study was authorized by the Consellería de Educación, Investigación, Cultura y Deportes (Regional Ministry) (CN00A/2018/25/S), the ethics committee at the Cardenal Herrera-CEU University (CEI18/112), and the Castellón Provincial Hospital Research Commission (CHPC-18/12/2019).

## 3. Results

A total of 3% of the adolescents (*n* = 12) presented with a DD compared with 16.4% (*n* = 65) who presented with PUVG. Regarding the sociodemographic characteristics of the cohort (Table 1), 66.7% of the adolescents with a DD were male. In terms of academic performance, 10% of the patients with a DD presented a mean grade corresponding to course failure, while this result only appeared in 0.8% of the healthy patients. In relation to the personality analysis according to the BFQ-NA (Table 2), we found that the mean conscientiousness, openness, and emotional stability dimension scores were significantly lower in adolescents with a DD compared with the group of healthy adolescents, while lower mean openness scores were obtained in adolescents with PUVG. Regarding parental socialization, according to the TXP questionnaire (Table 3), the affection–communication variable was significantly lower for the group of adolescents with a DD.

Our analysis of the study variables (Table 4 and Table 5) showed that they were all significantly correlated with the diagnostic variable, except for family living arrangements. The adjusted main effects multinomial regression (Table 6a) revealed that male gender best predicted a DD and a PUVG diagnosis, while scores indicating high emotional instability best predicted DDs. The unadjusted individual multinomial regressions (Table 6b) revealed that there was a significant relationship between DDs and school year failure or obtaining a pass/good designation. There was a significant relationship between conscientiousness, openness, and agreeableness with both PUVG and DDs. Regression analysis of the affection–communication variable showed a significant relationship both with PUVG and with DDs.

## 4. Discussion

This study investigated an area of growing scientific and clinical interest, namely, the presence of comorbid mental disorders in young subjects, specifically in the addictions research area. There is little literature regarding DDs in adolescents integrating behavioural pathologies, such as PUVG, together with substance addictions [30,31]. Therefore, this study aimed to explore the presence of the comorbid presence of gaming with other mental disorders within a population base sample of adolescents. A total of 3% of the adolescents who participated in this study presented a DD compared with 16.4% who presented PUVG. When we considered gender, we concluded that two-thirds of the adolescents with a DD were male. This finding agreed with the most recent scientific evidence [32], indicating that PUVG was more prevalent (72.3%) in males. Furthermore, male gender behaved as a predictive factor, both for PUVG and DDs. One possible explanation is that the dopaminergic reward system in the brain is more activated in men while playing video games compared with women [33].

Considering the family socialization model according to the TXP-A questionnaire [28], our study showed that providing affection and establishing communication with adolescents acts as a protective factor against the appearance of DDs and PUVG. Our data reinforced the hypothesis that the family socialization pattern is a determining factor in the appearance of DDs, while the majority of previous publications only related this factor to addictions [34]. Other studies showed that the absence of family affection–communication is generally related to psychopathology, and specifically, to conduct disorders and antisocial personality disorder [35].

On the other hand, considering personality by applying the BFQ-NA questionnaire, our study showed that low scores for conscientiousness, agreeableness, and openness predicted the risk of a DD and the onset of PUVG. In line with our results, the personality domains most often related to DDs in the literature are low conscientiousness and agreeableness, although, unlike our findings, no relationship with openness was found in previous work [36,37]. The same occurred for PUVG and GD [38,39], which was also associated with both low levels of conscientiousness and agreeableness and with male gender [40].

Focusing on conscientiousness, previous studies emphasized that this factor is not only lower in substance addicts but also in those with behavioural addictions, such as Internet use, gambling [41], and GD [42]. This would imply that these individuals have difficulty in complying with rules, lack autonomy, show less harm avoidance behaviour, tend to project blame onto third parties, and demonstrate disorder and imprecision, which would all act as an ideal combination for cultivating the start of a DD [36]. In terms of agreeableness, participants with a DD showed a greater tendency toward independence, a lower degree of social cooperation, and poor sensitivity to social stimuli [37], which are factors that would make outpatient follow-up difficult. In addition, our study showed that adolescents with a DD showed a greater tendency toward emotional instability (neuroticism).

According to recent academic literature, the personality trait that best differentiates individuals with a DD from those who only suffer from substance addiction is emotional instability [36]. If we analyze the role of personality in the genesis of addiction, neurotic individuals were described as more prone to low mood, with addiction tending to be their escape route from this sadness [43]. Therefore, it was surprising that our results did not show a correlation between PUVG and neuroticism, especially given that previous studies indicated that the use of video games acts as an avoidance or escape strategy in these individuals when facing stressful daily situations and exhaustion derived from emotional instability itself [44]. Previous publications showed that addressing the anxiety-depressive symptoms included in the emotional stability dimension is beneficial for the maintenance of substance abstinence [45].

Regarding academic performance, our results indicated that the presence of a DD would prevent adolescents from achieving their highest possible school grades, thereby making them more likely to fail. However, although research published on video games prior to the COVID-19 pandemic was also related to poor school performance [46], other articles defended the existence of an optimal gaming profile related to well-being, which would have potentially positive effects on these individual’s school environments [47]. Along this line, according to our data, PUVG alone was not a relevant factor in school performance, with the presence of psychopathology together with addiction being required for this relationship to become significant.

Of note, the COVID-19 pandemic significantly increased the use of video games among young adolescents [48]. However, several organizations also promoted the use of video games as an effective method to help people cope with the mental health challenges derived from the COVID-19 pandemic and the associated restrictive measures [49]. Notwithstanding, increased Internet gaming disorder severity was noted throughout the COVID-19 pandemic [48]. Therefore, it is essential to balance the time people spend engaging with video games (especially in more vulnerable populations, such as adolescents) by adopting specific preventive initiatives that help to curb addictive disorders related to video game use [50]. In addition, recent results suggest that symptoms of depression and anxiety during the pre-COVID-19 period positively predicted PUVG and GD during the pandemic [48].

Finally, it is worth mentioning the limitations of this study. The cross-sectional design of this work did not allow us to investigate the causal relationships between personality traits, gender, and family dynamics with the appearance of a DD. In addition, the study was based on self-reported and partially retrospective data collected in questionnaires administered to adolescents; therefore, our data may have been subject to recall bias and the diagnoses cannot be considered clinical. At this point, the BASC scale that is used to screen psychiatric disorders in adolescents is an insufficient diagnostic tool, especially in small samples, for which it would have been more appropriate to interview the patients that were found to be diagnosed with dual diagnoses. Another limitation was the small sample size of the dual diagnosis group, which could lead to false positives and negatives. The same occurred with regard to correlations with disorders: the collected estimates of odds ratios, confidence intervals, and *p*-values may have been biased due to the low representativeness of the sample (*p*-values of 0.04 and 0.06 were not far off in this respect and still led to a binary choice as to whether the item should be associated with pathology). Unfortunately, these limitations are common to most neuroscience studies and are alleviated by promoting replicability and methodological rigor [51]. Moreover, the term PUVG does not have specific standardized criteria unlike the terms IGD in the DSM-5 and GD in the ICD-11 used by other authors [52], although it would apply to those people that displayed excessive use of video games without reaching a significant functional impairment that would allow it to be defined as a video game addiction or gaming disorder, as has been described in previous research [53].

To date, drug addictions have enjoyed great hegemony within the field of DD, leaving behavioral addictions aside. In view of our results, and as also seen in recent scientific articles [54], all addictions should be considered as such, regardless of their toxic or behavioral natures. In addition, we must also consider whether a DD coexists with another mental disorder because this can significantly worsen the prognosis [55]. Treatment of these individuals is especially difficult because of their intrinsic characteristics and often dysfunctional parenting styles. Therefore, there is a strong consensus on the need to implement an integrated treatment program for adolescents with DDs [56] that would emphasize early detection and preventive interventions and consider the vulnerability of specific subpopulations by studying biological and environmental variations that contribute to their appearance [57]. Nowadays, there are inaccurate treatment protocols for IGD. Taking into account the functional impairment of the dopamine system, dopamine reuptake inhibitors, such as bupropion, showed good results in GD [58]. Non-pharmacological interventions, such as cognitive behavioral therapy, were also found to be effective in recent literature [58]. There have been no randomized clinical trials that considered a pharmacological treatment in GD dual disorders, but extrapolating from the importance of gambling dual disorders, a pharmacological approach should consider the perspective of clinical neuroscience and precision psychiatry [59].

In summary, the presence of an addiction to video games or substances in adolescents should be interpreted as a key red flag in these individuals because these factors themselves increase the risk of psychopathology. Thus, appropriate measures of primary prevention and early diagnosis to help avoid the appearance of a DD must be implemented as soon as possible because their emergence is associated with exponentially poorer patient prognoses.

## 5. Conclusions

The present study suggested that male gender; certain personality traits, such as low conscientiousness, openness, and kindness; and family dynamics with little affection and communication were associated with the presence of dual diagnosis and problematic use of video games in a sample of adolescents. In addition, emotional instability and poor academic results increased the risk of dual diagnosis. Therefore, it is necessary to implement programs for the early detection of these vulnerability factors for early intervention in the lives of adolescents and their families to prevent the appearance of dual diagnosis, where a dual diagnosis can not only be due to the coexistence of a mental disorder together with an addiction to substances but instead can apply to any type of behavioural addiction.

## Figures and Tables

**Figure 1 brainsci-12-01110-f001:**
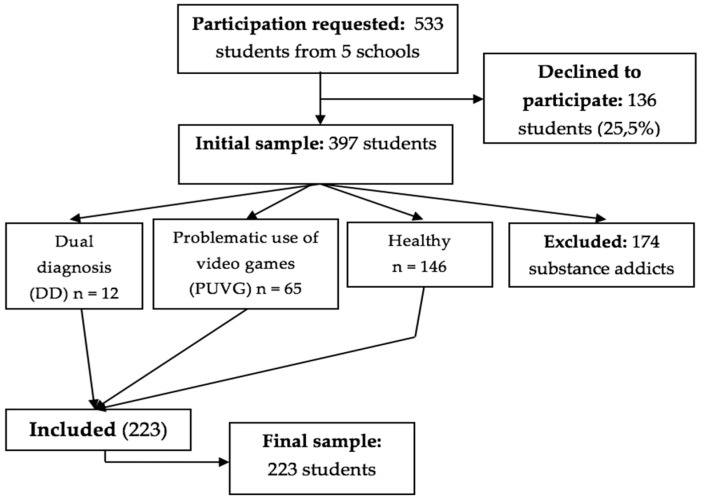
Sampling flowchart.

**Table 1 brainsci-12-01110-t001:** Sociodemographic characteristics of the adolescents included in this study (*N* = 223).

	Healthy	PUVG	DD	Statistics
**Female gender**	75.2% (*n* = 109)	27.7% (*n* = 18)	33.3% (*n* = 4)	χ^2^ 45.287 (*p* < 0.001) *V* 0.45 CTR H 6.7/CTR DD −6.1.
**Male gender**	24.8% (*n* = 36)	72.3% (*n* = 47)	66.7% (*n* = 8)	X^2^ 45.287 (*p* < 0.001) *V* 0.45 CTR H −6.7/CTR DD 6.1.
**Age in years**	M (SD) = 14.67 (0.69)	M (SD) = 14.60 (0.67)	M (SD) = 14.66 (0.70)	F 0.184 (*p* = 0.832) ES 0.002
**Number of siblings**	M (SD) = 1.97 (0.84)	M (SD) = 2.17 (1.08)	M (SD) = 2.11 (0.78)	F 0.949 (*p* = 0.389) ES 0.01
**Third year of compulsory secondary education**	48.3% (*n* = 70)	52.3% (*n* = 34)	58.3% (*n* = 7)	χ^2^ 0.644 (*p* = 0.725) V 0.05
**Fourth year of compulsory secondary education**	51.7% (*n* = 75)	47.7% (*n* = 31)	41.7% (*n* = 5)	χ^2^ 0.644 (*p* = 0.725) V 0.05
**Secular center**	67.8% (*n* = 99)	69.2% (*n* = 45)	91.7% (*n* = 11)	χ^2^ 2.981 (*p* = 0.225) V 0.11
**Catholic center**	32.2% (*n* = 47)	30.8% (*n* = 20)	8.3% (*n* = 1)	χ^2^ 2.981 (*p* = 0.225) V 0.11
**Private center**	34.2% (*n* = 50)	40.0% (*n* = 26)	41.7% (*n* = 5)	χ^2^ 3.397 (*p* = 0.494) V 0.08
**Chartered (state-subsidised) center**	28.8% (*n* = 42)	21.5% (*n* = 14)	8.3% (*n* = 1)	χ^2^ 3.397 (*p* = 0.494) V 0.08
**Public center**	37.0% (*n* = 54)	38.5% (*n* = 25)	50.0% (*n* = 6)	χ^2^ 3.397 (*p* = 0.494) V 0.08
**Repeated year: no**	87.8% (*n* = 108)	91.1% (*n* = 51)	90.9% (*n* = 10)	χ^2^ 0.463 (*p* = 0.793) V 0.04
**Repeated year: yes**	12.2% (*n* = 15)	8.9% (*n* = 5)	9.1% (*n* = 1)	χ^2^ 0.463 (*p* = 0.793) V 0.04
**Mean grade: equivalent to a ‘fail’**	0.8% (*n* = 1)	1.8% (*n* = 1)	10.0% (*n* = 1)	χ^2^ 16.106 (*p* = 0.003) V 0.20 CTR DD 2.2
**Mean grade: equivalent to a ‘pass/good’**	31.1% (*n* = 38)	36.4% (*n* = 20)	80.0% (*n* = 8)	χ^2^ 16.106 (*p* = 0.003) V 0.20 CTR DD 3.0
**Mean grade: equivalent to ‘remarkable / outstanding’**	68% (*n* = 83)	61.8% (*n* = 34)	10.0% (*n* = 1)	χ^2^ 16.106 (*p* = 0.003) V 0.20 CTR DD −3.6

M—mean; F—ANOVA statistic; CTR—corrected typified residuals; V—Cramer’s V (effect size of chi-squared); ES—effect size (partial eta-squared); n—number of participants; PUVG—problematic use of video games; DD—dual diagnosis; significant results are shown in bold.

**Table 2 brainsci-12-01110-t002:** Personality characteristics of the adolescents included in the study (*N* = 223).

	Healthy	PUVG	DD	Statistics: *F* (*p*) Tukey’s H*SD* (*p*)
	M (SD)	M (SD)	M (SD)	
**Conscientiousness (BFQ)**	57.19 (8.20)	52.57 (9.50)	45.77 (11.76)	F 10.537 (< 0.001) ES 0.10 H > PUVG (0.004) H > DD (0.001)
**Openness (BFQ)**	58.78 (8.94)	56.41 (9.65)	49.66 (10.44)	F 4.704 (0.01) ES 0.05 H > DD (0.014)
**Extraversion (BFQ)**	51.28 (9.45)	50.21 (9.81)	48.55 (14.14)	F 0.475 (0.623) ES 0.005
**Agreeableness (BFQ)**	54.24 (8.99)	50.30 (10.47)	49.44 (11.69)	F 3.723 (0.026) ES 0.04 H > PUVG (0.034)
**Emotional stability (BFQ)**	46.42 (11.08)	48.10 (9.81)	62.44 (10.19)	F 9.451 (< 0.001) ES 0.09 S < DD (< 0.001) PUVG < DD (0.001)

M—mean; SD—standard deviation; PUVG—problematic use of video games; DD—dual diagnosis; F—ANOVA statistic; ES—effect size (partial eta-squared); BFQ—Big Five Questionnaire; significant results are shown in bold.

**Table 3 brainsci-12-01110-t003:** The family dynamics of the adolescents included in the study (*N* = 223).

	Healthy	PUVG	DD	Statistics
**Living with both parents**	76.2% (*n* = 93)	83.9% (*n* = 47)	63.6% (*n* = 7)	χ^2^ 12.035 (*p* = 0.017) V 0.17
**Living with one parent alone**	23.8% (*n* = 29)	10.7% (*n* = 6)	27.3% (*n* = 3)	χ^2^ 12.035 (*p* = 0.017) V 0.17 CTR PUVG −2.1
**Other cohabitants**	0.0% (*n* = 0)	5.4% (*n* = 3)	9.1% (*n* = 1)	χ^2^ 12.035 (*p* = 0.017) V 0.17 CTR H −2.7/CTR PUVG 2.0
**Affection–communication (TXP)**	M (SD) = 87.75 (12.18)	M (SD) = 82.67 (12.19)	M (SD) = 73.66 (13.02)	F 7.642 (*p* = 0.001) ES 0.08 Tukey’s HSD (*p*) H > PUVG 0.032) H > DD (0.003)
**Control and structure (TXP)**	M (SD) = 35.47 (5.26)	M (SD) = 35.44 (4.95)	M (SD) = 36.77 (4.23)	F 0.281 (*p* = 0.755) ES 0.003

CTR—corrected typified residuals; n—number of participants; M—mean; SD—standard deviation; PUVG—problematic use of video games; DD—dual diagnosis; F—ANOVA statistic; ES—effect size (partial eta-squared); V—Cramer’s V (effect size of chi-squared); TXP—TXP parenting questionnaire; significant results are shown in bold.

**Table 4 brainsci-12-01110-t004:** Parametric variable correlations (*N* = 223).

	D	C	Op	Ag	ES	AC
**D**						
**C**	r = −0.361 *p* < 0.001					
**Op**	r = −0.231 *p* < 0.001	r = 0.748 *p* < 0.001				
**Ag**	r = −0.249 *p* < 0.001	r = 0.633 *p* < 0.001	r = 0.437 *p* < 0.001			
**ES**	r = 0.239 *p* < 0.001	r = −0.177 *p* < 0.001	r = −0.209 *p* < 0.001	r = −0.150 *p* = 0.003		
**AC**	r = −0.290 *p* < 0.001	r = 0.314 *p* < 0.001	r = 0.265 *p* < 0.001	r = 0.261 *p* < 0.001	r = −0.322 *p* < 0.001	

D—diagnosis; C—conscientiousness (BFQ); Op—openness (BFQ); Ag—agreeableness (BFQ); ES—emotional stability (BFQ); AC—affection–communication (TXP); r—Pearson’s correlation coefficient.

**Table 5 brainsci-12-01110-t005:** Nonparametric variable correlations (*N* = 223).

	D	G	MG	LA
D				
G	r = −0.442 *p* < 0.001			
MG	r = −0.180 *p* = 0.014	r = 0.023 *p* = 0.669		
LA	r = −0.016 *p* = 0.830	r = 0.003 *p* = 0.956	r = −0.066 *p* = 0.230	

r—Spearman’s correlation coefficient; significant results are shown in bold. D—diagnosis; G—gender; MG—mean grade; LA—living arrangements.

**Table 6 brainsci-12-01110-t006:** Adjusted (**a**) and unadjusted (**b**) multinomial regressions regarding problematic videogame use and dual diagnosis.

(a) Adjusted Multinomial Main Effects Regression		
	Problematic Use of Video Games OR [95% CI], *p*	Dual Diagnosis OR [95% CI], *p*
Male gender	9.854 [4.084–23.779], < 0.001	7.119 [1.132–44.785], 0.036
Emotional stability (BFQ)	1.017 [0.977–1.059], 0.412	1.116 [1.030–1.209], 0.008
**(b) Unadjusted Individual Multinomial Regressions**		
	**Problematic Use of Video Games** **OR [95% CI], *p***	**Dual Diagnosis** **OR [95% CI], *p***
Mean grade equivalent to a ‘fail’	2.441 [0.148–40.160], 0.532	83,000 [2.766–2490.921], 0.011
Mean grade equivalent to a ‘pass/good’	1.285 [0.656–2.517], 0.465	17,474 [2.110–144.705], 0.008
Conscientiousness (BFQ)	0.936 [0.905–0.969], < 0.001	0.843 [0.776–0.916], < 0.001
Openness (BFQ)	0.968 [0.939–0.999], 0.045	0.901 [0.843–0.962], 0.002
Agreeableness (BFQ)	0.951 [0.921–0.982], 0.002	0.920 [0.861–0.983], 0.013
Affection–communication (TXP)	0.968 [0.945–0.992], 0.009	0.927 [0.891–0.965], < 0.001

OR—odds ratio; 95% CI—95% confidence interval; BFQ—Big Five Questionnaire; TXP—TXP Parenting Questionnaire.

## Data Availability

The datasets used and analyzed during the current study are available from the corresponding author on reasonable request.

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
