# Peer review of "Dual Diagnosis in Adolescents with Problematic Use of Video Games: Beyond Substances"

_brainsci, 2022, doi:10.3390/brainsci12081110_

Round 1
Reviewer 1 Report
This is very interesting research and the topic is very important
Reviewer 2 Report
Dear Authors,
When I saw that your manuscript was submitted to the "Brain Sciences" journal, I expected that it would contain information relevant to neuroscience. Sadly, I do not find that your manuscript contributes to the field.
In addition, I conclude that your manuscript includes a few flaws that cannot be easily fixed through rewriting. Firstly, the number of the participants with dual diagnosis is too small to formulate any substantial conclusions, especially if based on p-values. Secondly, as some of the analyses are based on subgroups of n = 8, or even fewer participants, the findings are doubtful as are the subject to being influenced by one or two relative extreme values. In consequence, the analyses could have led to false positive/negative results.
Moreover, the selected set of variables does not seem to form a comprehensive theorizing to backup your exploratory analyses. I can imagine, that formulating hypothesis may not always be possible, but, still, the variables you tested seem to be a part of a pretty random set of psychological constructs which may have or may not have any meaning in predicting or treating problematic use of video games.
Additionally, effect sizes are not reported so it is not possible to evaluate how strong were the effect you found. Some tables include errors, e.g., incorrect total N is reported, Table 4 is hard to read due to formatting error; why there is the link between exclude participants and final sample (Figure 1). Lastly, it would good to know how the analysis correspond with the study goal: a guidance seems necessary as many analyses were conducted and reported in a relatively brief manuscript.
Given that, I cannot recommend your manuscript for printing in the journal, sadly. I suggest rejecting it.
Good luck with your projects.
Reviewer 3 Report
Few comments:
1. Line 47, "addictive-like behaviors". Did you mean to write "addiction-like behaviors"?
2. Line 67, "problematic use of video game use", please delete "use" after video game.
3. Line 310." to explore the presence of the comorbid presence of gaming with substance related problems." However, subjects with addiction to substances were excluded or not included in the analysis. Please clarify your point. My understanding is goal of your study is to explore the association of variables with DD and problematic use of video games excluding substance use.
Round 2
Reviewer 2 Report
Dear Authors,
Unfortunately, I cannot recommend publishing the manuscript in the journal. There are two reasons for it.
Firstly, the manuscript does not contribute to neuroscience.
Secondly, the group with dual diagnosis is too small. I do not see any reason for multiplying such a major and irreversible flaw in another publication.
Besides, I want to address another minor problems.
(a) I still cannot identify a comprehensive theoretical justification for the tested set of variables. Indeed, you have amended the introduction with a new reference, but, still, it is not an apt conceptual framework.
(b) Effect size shall be reported regardless the level of p-value. We report effect size if a p-value is lower or higher than 0.05.
(c) The text in lines 222-258 is ill-formatted, and thus, hard to read.
(d) Table 4 includes duplicated data (above the diagonal is the same information as below it).
All the best.
Author Response
Unfortunately, I cannot recommend publishing the manuscript in the journal. There are two reasons for it. Firstly, the manuscript does not contribute to neuroscience. Secondly, the group with dual diagnosis is too small. I do not see any reason for multiplying such a major and irreversible flaw in another publication.
The authors respect your opinion, and we continue to be grateful for the opportunity you give us to continue improving the manuscript.
Besides, I want to address other minor problems:
(a) I still cannot identify a comprehensive theoretical justification for the tested set of variables. Indeed, you have amended the introduction with a new reference, but, still, it is not an apt conceptual framework.
We have dedicated part of the introduction to a deeper description of the concept of "dual diagnosis" as a theoretical justification to help the reader understand the rationale for the variables used in the study. We have added the following sentences:
- Lines 47-50: Problematic Internet use includes diverse activities apart from videogames, as social media use, web-streaming, buying and pornography viewing, which have been correlated with emotional dysregulation (Pettorruso, 2020)
- Lines 71-77: The pandemic of COVID-19, lockdown and limitation of movements imposed by authorities increased the overall use of internet and videogames (Gioneska, 2021). Moreover, rates of psychopathology (depression, anxiety, post-traumatic stress symptoms) in patients with substance or behavioral addiction have increased moderately, resulting in a poor quality of life (Martinotti, 2020). In this context, the recently described term cyberchondria understood as excessive online searching for medical information must be considered, taking into account the elevated use of internet (Vismara, 2020)
- Lines 80-84: At this point, the way of understanding and dealing with addictions is changing, because previously only substance disorders were considered. The underlying reason is the strong correlation between playing videogames and gambling, which as well, has commonalities in clinical expression, etiology, physiology and comorbidity with substance use disorders according to DSM-5 (Jimenez-Murcia, 2014)
(b) Effect size shall be reported regardless the level of p-value. We report effect size if a p-value is lower or higher than 0.05.
We have included all effect sizes in Tables 1, 2 and 3.
(c) The text in lines 222-258 is ill-formatted, and thus, hard to read.
Dear reviewer, we have requested the Editor to re-send the manuscript without errors in the format. The Table 4 gave us some errors, so the text next to that moved as well. What we have done is deleting and copying again the Table and adjust the text to the Table.
(d) Table 4 includes duplicated data (above the diagonal is the same information as below it).
It is true that the data in Table 4 and also in Table 5 was duplicated because it contained parametric and nonparametric variable correlations. Table 4 contains information about correlation of 6 variables: D = Diagnosis; C = Conscientiousness; Op = openness; Ag = agreeableness; ES = emotional stability and AC = affection-communication. So, the previous data had redundant information about the correlation: for example, it appeared the correlation between Conscientiousness and Openness and also between Openness and Conscientiousness. As the values are exactly the same, we have deleted the data above the diagonal in both Tables (4 and 5), so it is more visual and non-redundant. Thank you for your comments.
All the best.
Thank you very much for your encouragement.
